# Cytomorphology of Noninvasive Follicular Thyroid Neoplasm with Papillary-Like Nuclear Features and the Impact of New Nomenclature on Molecular Testing

**DOI:** 10.3390/medsci7020015

**Published:** 2019-01-22

**Authors:** Rupendra T. Shrestha, Darin Ruanpeng, James V. Hennessey

**Affiliations:** 1University of Minnesota Twin Cities, Department of Medicine, Minneapolis, MN 55455, USA; shres053@umn.edu (R.T.S.); ruanp001@umn.edu (D.R.); 2Division of Endocrinology, Beth Israel Deaconess Medical Center, 330 Brookline Avenue, GZ-6, Boston, MA 02215, USA

**Keywords:** NIFTP, non-invasive follicular thyroid neoplasm with papillary-like nuclear features, Bethesda, FVPTC, follicular variant of Papillary thyroid cancer, *RAS*, *BRAF*

## Abstract

The re-naming of noninvasive follicular variant papillary thyroid cancer to the apparently non-malignant, noninvasive follicular thyroid neoplasm with papillary-like nuclear features (NIFTP) impacts the prevalence of malignancy rates, thereby affecting mutation frequency in papillary thyroid cancer. Preoperative assessment of such nodules could affect management in the future. The original publications following the designation of the new nomenclature have been extensively reviewed. With the adoption of NIFTP terminology, a reduction in the follicular variant of papillary thyroid cancer (FVPTC) prevalence is anticipated, as is a modest reduction of papillary thyroid cancer (PTC) prevalence that would be distributed mainly across indeterminate thyroid nodules. Identifying NIFTP preoperatively remains challenging. *RAS* mutations are predominant but the presence of *BRAF* V600E mutation has been observed and could indicate inclusion of the classical PTC. The histological diagnosis of NIFTP to designate low-risk encapsulated follicular variant papillary thyroid cancers (EFVPTCs) would impact malignancy rates, thereby altering the mutation prevalence. The histopathologic criteria have recently been refined with an exclusion of well-formed papillae. The preoperative identification of NIFTP using cytomorphology and gene testing remains challenging.

## 1. Introduction

The rising incidence, stable prevalence and unchanged mortality of thyroid cancer have led many authors to conclude that we may be over-diagnosing and overtreating thyroid cancers [1,2,3]. A diagnosis of cancer has significant emotional and financial risks to the patients [4]. To address the issue of overdiagnosis, a National Cancer Institute conference was convened in 2012 which concluded that a more indolent term to define certain encapsulated follicular variant papillary thyroid cancers (EFVPTCs) was needed [5]. The Endocrine Pathology Society working group was created to address this task. This working group performed a retrospective study of 268 tumors diagnosed as EFVPTCs including mutation testing of a cohort of these tumors. A subset of low risk-encapsulated follicular variant papillary thyroid cancer was re-named noninvasive follicular thyroid neoplasm with papillary-like nuclear features (NIFTP), utilizing a set of strict diagnostic criteria (Table 1) [6]. In this category, encapsulated or clearly demarcated papillary thyroid cancers with predominant follicles are included with a nuclear score between two and three. The nuclear scores are based on: (1) size and shape (nuclear enlargement, overlapping, and/or elongation), (2) nuclear membrane irregularities (irregular contours, grooves, and/or pseudo-inclusions), and (3) chromatin characteristics (chromatin clearing, margination of chromatin to membrane, and/or glassy nuclei). Each class of nuclear features is assigned a score of zero or one, yielding a range of scores from zero to three. The exclusion criteria include the presence of psammoma bodies; papillae more than 1%; 30% or more solid/trabecular/insular growth pattern; capsular or vascular invasion; high mitotic activity, and presence of tumor necrosis. Such NIFTP tumors are expected to have an excellent prognosis based upon retrospective studies regardless of size [7], and therefore it is assumed that NIFTP patients would not require extensive surgery or radioactive iodine treatment.

## 2. Prevalence of Non-Invasive Follicular Thyroid Neoplasm with Papillary-like Nuclear Features (NIFTP)

A large majority of NIFTP have been reclassified from Follicular Variant of Papillary Thyroid Cancers (FVPTCs) and the prevalence of FVPTCs is rising [2]. The prevalence of follicular variant papillary thyroid cancer (PTC) is thought to vary from 22% to 43% of all papillary thyroid cancer (PTC) variants [6]. However, a prevalence lower than 5% has been reported [8,9]. The prevalence of NIFTP among FVPTCs varies widely from 17% to 71% in the literature since the publication of diagnostic criteria by Nikiforov et al. in 2016 [10,11]. The overall impact of reclassification, therefore, depends on the prevalence of FVPTCs but the impact in overall malignancy rate may be small [8,12,13,14]. Variability of rates may indicate diverse study populations, the inclusion of microcarcinomas or interobserver variability in the diagnosis of FVPTCs [15].

## 3. Alteration in Malignancy Rate with the Introduction of Non-Invasive Follicular Thyroid Neoplasm with Papillary-like Nuclear Features (NIFTP)

Since significant portions of noninvasive FVPTCs are categorized as atypia of indeterminate significance (AUS), suspicious for follicular neoplasm (SFN) and suspicious for malignancy (SUS), the greatest impact of this new nomenclature was expected in these categories [6,16,17]. Additionally, initial studies that included the noninvasive FVPTC showed a significant risk reduction of malignant lesions not only in all three indeterminate categories (AUS, SFN and SUS) but also in the benign category, thereby reducing the false positive results in cytology [16,17,18]. The greatest impact was seen in the SUS category of nearly 31–42% absolute risk reduction of malignancy with the reclassification [19,20,21]. Strict NIFTP criteria were not applied in all studies published prior to the standardization of such terminology and the studies indicate that only a portion of PTCs meets strict NIFTP criteria [8,22].

## 4. Impact of Non-Invasive Follicular Thyroid Neoplasm with Papillary-like Nuclear Neatures (NIFTP) on Bethesda Classification of Thyroid Cytology

The preoperative evaluation of NIFTP has been an area of great interest to cytopathologists. The identification of a potential NIFTP after a cytology review will help plan for discussions with the patient and support a more conservative surgery. Various studies have looked into the Bethesda categories where NIFTP has eventually been diagnosed to identify differences in cytological classification [8,10,12,13,19,20,21,23,24,25,26,27,28,29,30,31,32,33,34,35,36]. In the studies conducted in the US that have utilized contemporary NIFTP criteria, more than 80% of the diagnoses fall under the AUS, SFN and SUS categories [12,19,20,21,30,31,33]. By comparison, when assessed with cytology, infiltrative variants are more likely to be SUS or malignant [13]. While some have reported that a malignant preoperative cytologic diagnosis is less likely in NIFTP [24,34], others have shown a high rate of preoperative malignant cytology in the Bethesda category that corresponds to NIFTP [8]. To address the alteration of malignancy rates, the Bethesda classification now includes the expected risk of malignancy that accounts for alterations due to NIFTP [37].

## 5. Cytomorphology of Non-Invasive Follicular Thyroid Neoplasm with Papillary-like Nuclear Features (NIFTP)

The identification of cytomorphological features to distinguish NIFTP from its invasive counterpart as well as with classical PTC (cPTC) has been attempted in various studies. Compared to cytology of a classical PTC (cPTC), the noninvasive FVPTC or NIFTP cases (see Table 2) are more likely to have microfollicular predominant patterns, and are less likely to have papillae, pseudo-inclusion or sheet predominant patterns [18,38]. It has been further suggested that the presence of the nuclear features of PTC in a microfollicular pattern could suggest the presence of NIFTP [30]. Similarly, when compared to papillary thyroid carcinoma with predominant follicular pattern (PTC-FP), NIFTP cytology was more likely to be of a microfollicular pattern, lack pseudo-inclusion and present with a nuclear score of two [25]. Although a definitive diagnosis cannot be established on cytology, NIFTP lesions were less likely to have nuclear grooves and more likely to have smaller nuclear sizes [26]. While an increased nuclear size, membrane irregularities and chromatin clearing was more likely to be present in NIFTP compared to benign follicular lesions, no differences have been noted when these features are compared to invasive FVPTC (I-FVPTC) [27]. The identification of the differences in cytology between NIFTP and I-FVPTC remains challenging. While a single study found more nuclear folds in I-FVPTC [29], others have found no distinguishing features in cytology [29,39]. Larger studies are needed in this area but given the significant overlap with its invasive counterpart, the accurate determination of NIFTP in cytology may be difficult to impossible in general practice.

## 6. Mutational Testing in Non-Invasive Follicular Thyroid Neoplasm with Papillary-like Nuclear Features (NIFTP)

*RAS* mutations have been shown to be the predominant mutations detected in noninvasive FVPTCs, especially in lesions with indeterminate cytology [40,41] and therefore, recategorization is set to alter the mutational landscape of PTCs. Following consensus on NIFTP nomenclature, several papers have reported the mutational testing results of this category. *BRAF* V600E, the most common driver mutation in PTC [42], is uncommon in NIFTP tumors. *BRAF* V600E mutations have been absent in NIFTP nodules in several studies [22,30,31] but other variants of *BRAF* such as K601E, T599_R603 and V600M have been reported in a minority of nodules [10,31,43,44]. However, the presence of *BRAF* V600E mutations may correlate with papillary structures in NIFTP and may therefore represent an overlap with classical PTC. When all tumors with papillae were excluded from NIFTP, none of the remaining EFVPTCs had *BRAF* mutations [27,45]. On the other hand, *RAS* mutations are the predominant mutations in NIFTP. Among 27 *RAS*-mutated thyroid tumors, 59% (16/27) met NIFTP criteria. [40]. *NRAS* and *HRAS* mutations were more common compared to *KRAS* mutations and multiple mutations have been noted. [44] Similarly, nearly half of the NIFTPs that were tested for the multigene panel were positive for variants of *RAS* mutations [36,45]. Other occasionally reported mutations include *TERT* mutation [35], *PAX8*-*PPARG* and *CREB3L2*-*PPARG* fusions [36] and *THADA* fusion [44]. Assuming the trend remains, *RAS* mutations in the remaining PTCs are expected to fall whereas the remaining classic PTCs and the invasive variants are more likely to harbor *BRAF* V600E mutations.

A few studies have reported the results of the Gene Expression Classifier (GEC) testing in indeterminate thyroid nodules [11,35,44,46,47,48,49] which were reported to be NIFTPs on final histopathology. The majority of NIFTP lesions in these studies were categorized as suspicious on GEC testing. In a study by Song et al., 26 out of 32 of the NIFTP cases that underwent GEC testing showed a suspicious result [35]. Therefore, GEC testing cannot distinguish NIFTPs from PTCs and would most likely identify the lesions as suspicious prompting a hemithyroidectomy. Consequently, the positive predictive value of this test is expected to lower further if NIFTP is considered nonmalignant, but the negative predictive value of the test would potentially increase. Recently, GEC testing has been replaced by Genomic Sequencing Classifier (GSC) which includes testing for *BRAF* mutations for suspicious lesions along with an Expression Atlas that can test for *RAS* mutations. A positive *RAS* mutation in such lesions may raise the possibility of NIFTP in appropriate contexts in GSC suspicious lesions.

The entire premise of utilizing the diagnostic category of NIFTP is to reduce the overdiagnosis and overtreatment of tumors with low malignancy potential and to guide long-term treatment. NIFTP is a histological diagnosis and while certain cytomorphological features may be helpful to suspect the presence of NIFTP, a preoperative diagnosis based on cytology is not possible using the Bethesda system. Despite the use of strict criteria, micrometastasis to the lymph node may occur. In a study of 154 encapsulated FVPTCs, when a cutoff of 1% papilla was used, the lymph node metastasis rate was 3% and the presence of *BRAF* V600E mutations was seen in 10% of the tumors. In the same study, no *BRAF* mutations were noted when the absence of papillae was used as a cutoff, but lymph node micrometastasis still occurred in 3% of the non-invasive tumors. This indicates that NIFTP may not be a completely benign entity and may represent a non-homogenous group of tumors that appear alike in histopathology [45]. So far, molecular testing has not shown a homogenous distribution of the mutations in NIFTP tumors. Based on the studies published [9,27,45], the criteria for NIFTP have now been updated and should no longer include any lesions with well-formed papillae or high risk *BRAF, TERT* or *TP53* mutations [50].

## 7. Future Directions

The PD-L1 biomarker has been evaluated for use in distinguishing NIFTP from invasive EFVPTCs. In a study of 174 tissue blocks of surgically removed thyroid nodules, cytoplasmic PD-L1 expression was significantly increased in the invasive forms compared to the NIFTP [51]. However, only about 6% of PTCs stain for PD-L1 [52]. Hector Battifora mesothelial-1 (HBME-1), cytokeratin-19 (CK19), galectin-3 (Gal-3), and CD56 expression have been studied in cell-blocks of follicular-patterned tumors and a scoring system to differentiate the infiltrative forms from the encapsulated forms has been attempted [53] with some success. However, immunohistochemical staining was not successful in identifying NIFTP from invasive FVPTCs. Further studies are needed to find more cost-effective ways to distinguish NIFTPs preoperatively.

## 8. Conclusions

The use of NIFTP to designate a clinically low-risk EFVPTC impacts malignancy rates in the indeterminate thyroid cytology. A great majority of these tumors have *RAS* mutations, but some may carry *BRAF* V600E mutations. The criteria for NIFTP identification have recently been refined with the exclusion of well-formed papillae and high-risk mutations. The preoperative identification of NIFTP using cytomorphology and gene testing remains challenging and therefore NIFTP continues to be a histopathologic diagnosis.

## Figures and Tables

**Table 1 medsci-07-00015-t001:** Noninvasive follicular thyroid neoplasm with papillary-like nuclear features (NIFTP) criteria.

	Criteria
1	Encapsulation or clear demarcation
2	Nuclear score 2–3
3	No vascular or capsular invasion
4	No tumor necrosis
5	No high mitotic activity (<3/HPF)
6	Follicular growth pattern with: <1% Papillae (criteria modified in 2018 to “no well-formed papillae”) No psammoma bodies <30% solid/trabecular/insular growth pattern

HPF: high-power fields.

**Table 2 medsci-07-00015-t002:** Summary of the cytomorphological differences between NIFTP and classical papillary thyroid cancer (cPTC).

Cytomorphology	NIFTP	PTC	*p*-Value
Architectural feature			
Bizzarro et al. (Italy, 2017) [26]			
- Presence of papillae	0/37	40/40	<0.00001
- Presence of isolate cells	8/37	34/40	<0.00001
- Presence of molding arrangement	17/37	31/40	0.00525
- Median colloid globules	3.1	2.3	>0.05
Brandler et al. (USA, 2017) [30]			
- Presence of papillae	3/56	47/67	<0.001
- Presence of abundant colloid	16/56	7/67	0.02
- Calcification	2/56	15/67	<0.01
- Microfollicle	41/56	2/67	<0.01
Strickland et al. (USA, 2016) [18]			
- Presence of papillae	1/8	30/42	<0.0001
- Psammomatous calcification	0/8	7/42	0.179
- Micro follicle	5/8	2/42	<0.0001
Howitt et al. (USA, 2015) [38]			
- Presence of papillae	0/11	14/28	0.0030
- Microfollicle	6/11	1/28	0.0009
- Sheet predominant	4/11	27/28	0.0002
Diaz et al. (Spain, 2018) [29]			
- Presence of papillae	1/6	13/14	0.001
- Psammomatous calcification	0/6	0/14	N/A
- Microfollicle	6/6	8/14	NS
- Dirty background	2/6	3/14	NS
- Tridimensional group	3/6	14/14	0.014
Jaconi et al. (Italy, 2017) [39]			
- Presence of papillae	0/14	20/30	N/A
- Psammomatous calcification	3/14	18/30	N/A
- Microfollicle	11/14	2/30	N/A
Nuclear feature			
Bizzarro et al. (Italy, 2017) [26]	11/37	36/40	<0.00001
- Size >20 um	6/37	38/40	<0.00001
- Presence of pseudo-inclusion	5/37	40/40	<0.00001
- Presence of groove	9/37	0/40	0.00077
- Regular nuclear membrane			
Brandler et al. (USA, 2017) [30]			
- Nuclear enlargement	47/56	66/67	<0.01
- Presence of pseudo-inclusion	5/56	58/67	<0.001
- Presence of groove	20/56	59/67	<0.001
- Nuclear irregularity	6/56	31/67	<0.001
- Nuclear crowding	46/56	66/67	<0.01
- Nuclear clearing/washout/powdering chromatin	39/56	65/67	<0.001
Strickland et al. (USA, 2016) [18]			
- Presence of pseudo-inclusion			
Howitt et al. (USA, 2015) [38]	1/8	35/42	<0.0001
- Presence of pseudo-inclusion			
Diaz et al. (Spain, 2018) [29]	0/11	22/28	<0.0001
- Presence of pseudo-inclusion			
- Presence of groove	5/6	13/13	NS
- Nuclear clearing	4/6	13/13	0.004
Jaconi et al. (Italy, 2017) [39]	5/6	10/13	NS
- Presence of pseudo-inclusion			
- Irregular nuclear/nuclear groove	0/14	22/23	N/A
- Nuclear enlargement/crowding	4/14	27/30	N/A
- Nuclear clearing/washout	10/14	30/30	N/A
	4/14	25/30	N/A
Other			
Brandler et al. (USA, 2017) [30]			
- Giant cell	4/56	28/67	<0.001
Diaz et al. (Spain, 2018) [29]			
- Giant cell	0/6	7/14	0.032
Jaconi et al. (Italy, 2017) [39]			
- Giant cell	2/14	25/30	N/A

NS—Non-significant (*p* > 0.05); N/A—not applicable; PTC—papillary thyroid cancer.

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
