# Peer review of "Cytomorphology of Noninvasive Follicular Thyroid Neoplasm with Papillary-Like Nuclear Features and the Impact of New Nomenclature on Molecular Testing"

_medsci, 2019, doi:10.3390/medsci7020015_

Round 1
Reviewer 1 Report
This is a very good paper dealing with a substantial problem in thyroid fine needle biopsy. The paper contribues to a better understanding of NIFTP.
Author Response
Thank you so much for your comment.
Reviewer 2 Report
This is a well-written and well-organized paper summarizing histo- and cyto-morphology of NIFTP and its impact on the practice managing thyroid neoplasm.
Minor comment:
Regarding the prevalence of NIFTP and malignancy rate with introduction of NIFTP, authors should update the rates. A recent study (PMID: 29476382; Endocr Pathol. 2018; 29: 276–288) performed a meta-analysis and estimated the overall prevalence of NIFTP among PTCs or thyroid malignancies in different countries of non-Asian and Asian series. It is recommended to cite the references and their results.
Author Response
Thank you for the comment and suggestion.
We have added reference 14 on line 62 as suggested.
Reviewer 3 Report
Generally speaking, the manuscript is clearly, concisely written and their conclusions seem appropriate and rationale.
However, some minor changes should be performed:
Please check the few spelling grammatical errors in the text.
Author Response
Thank you for the comment and suggestion.
We have made grammatical check and corrected as attached.